# Patient Factors Associated with Different Hospice Programs in Korea: Analyzing Healthcare Big Data

**DOI:** 10.3390/ijerph19031566

**Published:** 2022-01-29

**Authors:** Young-Taek Park, Daekyun Kim, Su-Jin Koh, Yeon Sook Kim, Sang Mi Kim

**Affiliations:** 1HIRA Research Institute, Health Insurance Review & Assessment Service (HIRA), Wonju 26465, Korea; pyt0601@hira.or.kr; 2Department of Family Medicine, Incheon St. Mary’s Hospital, College of Medicine, The Catholic University of Korea, Incheon 21431, Korea; bloves@naver.com; 3Department of Hematology and Oncology, Ulsan University Hospital, Ulsan University College of Medicine, Ulsan 44033, Korea; sujinkoh@uuh.ulsan.kr; 4Department of Nursing, California State University San Bernardino, San Bernardino, CA 92407, USA; yeon.kim@csusb.edu; 5Department of Health Management, Jeonju University, Jeonju 55069, Korea

**Keywords:** hospice, palliative care, hospice shared care, hospice units, terminal illness

## Abstract

The Korean government has implemented a pilot project that introduces a new type of hospice care program called “Consultative Hospice Care” (COHC) since August 2017. The COHC is a new type of hospice program for terminally ill patients in acute care wards, which is different from the Independent Hospice Unit (IHU) care. This study aimed to compare the characteristics of two groups of hospice patients: COHC care only and both IHU care and COHC groups. Healthcare claim data from 1 April 2018 to 31 March 2020 were retrieved from the HIRA data warehouse system. The main outcome variable was patients receiving COHC only or both COHC and IHU care. The total number of hospice patients was 6482. A multivariate logistic regression analysis was used. Of 6482 hospice care recipients, 3789 (58.5%) received both COHC and IHU care. Those who received both COHC and IHU care were significantly associated with several factors: period from the first evaluation to death (adjusted odds ratio (aOR), 1.026; 95% confidence internal (CI), 1.024–1.029; *p* < 0.0001), disease severity measured by the Charlson Comorbidity Index (aOR, 1.032; CI, 1.017–1.047; *p* < 0.0001), consciousness (aOR, 3.654; CI, 3.269–4.085; *p* < 0.0001), and awareness of end-stage disease (aOR, 1.422; CI, 1.226–1.650; *p* < 0.0001). The COHC program had a critical role in hospice delivery to terminally ill patients. Policymakers on hospice care need to establish plans that promote efficient hospice care delivery systems.

## 1. Introduction

The Ministry of Health and Welfare in South Korea has implemented a pilot project that introduces a new type of hospice care program called “consultation-based hospices” (“Consultative Hospice Care”, Hereafter “COHC”) since August 2017. The COHC has been known as “hospice shared care” in Taiwan [1,2]. The COHC is a new type of hospice program for terminally ill patients in an acute care unit, which is different from hospice care in an Independent Hospice Unit (IHU). In the COHC, hospice care team is providing hospice care and consultations to patients with terminal illness in acute care units [3,4].

In addition to the COHC, there are two other types of hospice programs in Korea. One is the hospice care provided in the IHU. The IHU is one of the hospital units that implement professional hospice care programs. This program was authorized to be covered by the national health insurance program in July 2015. The other is the hospice care provided at home or in community settings by the hospice team dispatched from a hospital. This program is generally called “home-based hospice care” [5,6] or “home hospice care” [7]. This program was authorized to be covered by the national health insurance program in September 2020.

According to the guideline booklet of the Korean government on the COHC pilot project [8], it was anticipated that the COHC program would help patients with terminal illness receive better end-of-life care by registering them to the IHU program earlier. In other words, the Korean government expected the COHC program to serve as the forefront gateway or bridge for early entry into IHU hospice care. This aim was also expected in the pilot project on home hospice care, which was applied as a nationwide program in 2020. According to a study analyzing the pilot project, the home hospice care program was effective in early enrollment of patients into the hospice care program [7]. As mentioned above, since COHC was used as a bridge to IHU hospice entry, patients using both COHC and IHU would have a longer stay period from entry to hospice to death compared to patients using only COHC. This longer period of stay in hospice care programs may suggest that both users be mentally and physically better than those using only COHC program at the time of hospice enrollment, and as a result, the degree of awareness on their end-stage diseases would be high. For mental and physical status of patients, this study selected three factors: disease severity, patient’s consciousness, and their awareness of end-stage disease. Investigating the relationships between use of different hospice programs and these factors is one of major features of this study. However, no previous study evaluated its relationships of the pilot project in these standpoints. Accordingly, it is necessary to evaluate whether early entry into the IHU was achieved and whether there were any relationships between use of different hospice programs such as COHC only or both hospice programs (COHC and IHU) and characteristics of patients in the standpoint of disease severity, patient’s consciousness, and their awareness of end-stage disease. This study hypothesized that hospice recipients who use both would have longer stays in the hospice program and better physical and mental status compared to those recipients with COHC only. For readers’ better understanding, this study constructed the presentation of the study results in the order of patients’ enrollment period of hospice program, which is the main purpose of pilot project followed by disease severity, patient’s consciousness, and their awareness of end-stage disease.

Regarding the period from the initial registration of hospice to death, a study conducted in 2020 showed that the percentage of patients with less than 7 days of hospice length of stay was highest in hospital-referred patients than those referred from any other location [9]. Thus, a hospice program, COHC, was introduced with a relatively short period to help early entry into hospice care, and patients receiving both COHC and IHU care would have a high possibility of longer length of stay compared to those receiving only COHC. 

For disease severity, a study conducted in Australia found that patients with cancer had died more in hospice care compared to those without cancer [10]. According to a hospice utilization study in the United States of America, the odds of receiving hospice care were associated with the presence of cancer [11]. The most prevalently observed primary diagnosis in hospice care was cancer [9]. These studies suggest that patients using both programs would be more likely severely ill. The Charlson Comorbidity Index (CCI) is used to evaluate disease severity, but a study conducted in Taiwan showed that the CCI had a limited role for severity evaluations of hospice care [12].

Regarding patients’ consciousness, a study conducted in 2016, which compared patients who were referred to hospice care more than 7 days before death, found that late referral (referral within 7 days before death) was associated with patient characteristics, such as “bedbound at admission”, “aphasic”, “unresponsive”, or “dyspneic” [13]. In a multivariable analysis, patients discharged to hospice care were older, had higher a National Institute of Health Stroke Scale score, and were present with altered mental status compared to those discharged to non-hospice care [14]. These study results indicate that referrals from COHC to IHU mean that both users may be related to patients’ mental status.

Regarding the awareness of end-stage disease, a study investigated the features of patients using hospice palliative care units and found that older age and awareness of terminal illness were positively associated with utilization of a hospice palliative care unit [15]. According to another study, patients who were aware of their terminal illness showed lower anxiety and depression scores and were more likely to sign the do not resuscitate consent than those who were unaware or partially aware [16,17]. These studies indirectly suggest that patients receiving both COHC and IHU care are more likely aware of end-stage disease.

This study aimed to investigate the relationships between the use of different hospice programs and the four characteristics of hospice care recipients: the enrollment period of hospice program, disease severity, patient’s consciousness, and their awareness of end-stage diseases.

## 2. Materials and Methods

### 2.1. Study Design

The units of analysis were individual patients who received COHC. This study had a cross-sectional study design using hospice utilization data between 1 April 2018 and 31 March 2020. There were 97 hospitals implementing any types of hospice programs as of 31 December 2020. Among them, 70 hospitals were providing hospice care with IHU, 9 were offering COHC only, and 18 hospitals had both programs with COHC and IHU. The main study was conducted in the Health Insurance Review and Assessment (HIRA) Service in Korea. For the study purpose, this study was approved by the Institutional Review Board on 14 April 2020 (IRB No: 2020-026-002).

### 2.2. Data Sources

Three main sources of data were used: HIRA, National Hospice Center, and Statistics Korea (Figure 1). Most data regarding demographic information, except the patient’s location of residence, were from the National Hospice Center, one of departments of the National Cancer Center in Korea. The National Hospice Center has the Korean Hospice Registry Database. Any patients who want to receive hospice care are obligated to submit their demographic data to the Korean Hospice Registry Database with the Case Report Form.

Information on date of death and residential location of patients was obtained from the National Statistics Korea. The rest of the information was obtained from HIRA, including data on IHU and disease severity. Table 1 presents the overall data sources. The national residents’ identification numbers were used for data linkage and data merge.

### 2.3. Outcome Variables and Independent Variables

The unique outcome variable of this study is whether hospice patients received only COHC or both COHC and IHU care. Although HIRA has all information on patients’ healthcare utilization and costs, it did not have any further detailed information on patient’s demographic information, such as main caregiver and living status with others. Therefore, this study mainly used patient data from the National Hospice Center. By collecting all data on COHC and linking them with data from HIRA’s main data warehouse systems, patients were grouped whether they received only COHC or both COHC and IHU care. The National Hospice Center data contained a Case Report Form including two types of information: registration and enrollment data. Patients who wanted to receive hospice care were supposed to complete initially the registration data and then the enrollment data whenever they were hospitalized in the hospice facility or received COHC.

This study had four target independent variables: period between the first hospice registration and death, disease severity of patients measured by the CCI, consciousness of patients at the time of the first registration, and awareness of end-stage disease. This information was from the Korean Hospice Registry Database of the National Hospice Center, except the CCI. Patients who wanted to enroll into the hospice program the first time or at the beginning of any hospice program should fill out the Case Report Form, and the information was sent to the Korean Hospice Registry Database.

For the disease severity of patients, this study used the CCI. The study retrieved all health insurance claims of the study subjects from HIRA’s data warehouse system for the last two years before the patient’s death, including hospice care, hospitals, and clinics. The diagnosis code of the claims used the seventh version of Korean Standard Classification of Diseases and Causes of Death, which is equal to the 10th extension version of the International Classification of Disease codes [18]. By using all diagnosis codes of patients, this study calculated the CCI score as it was conducted by Quan et al. [19].

The period between the first hospice registration and death means the number of days from the first registration of patients in the hospice care unit (COHC or IHU) to death. Consciousness is defined as a patient’s mental status and measured by four categories: “alert”, “drowsy”, “stupor”, and “coma”. In this study, it was classified into two categories: alert versus not alert. Awareness of end-stage disease indicates whether a patient is aware of end-stage disease, which is measured as “aware” or “not aware” of the disease. 

### 2.4. Statistical Analysis

The dependent variable was a binary scale: COHC only or both COHC and IHU (COHC only: 0 vs. both COHC and IHU: 1). By establishing this outcome variable, this study analyzed the general characteristics of the independent variables using cross-tabulation, chi-square test, and t-test of mean difference. Before conducting the main analysis, this study examined the correlations among independent variables to check the multicollinearity issue of independent variables. There was a high correlation among target independent variables, leading the study to establish separate models to consider this effect. This study used multivariate logistic regression and suggested 95% confidence intervals for each independent variable. SAS version 9.4 (SAS Institute Inc., North Carolina, NC, USA) was used for the data analysis.

## 3. Results

### 3.1. General Characteristics of the Study Subjects

Table 2 presents the general characteristics of the study subject. There were a total of 6482 patients who received COHC and IHU care in the records. Among them, 58.5% of patients received both COHC and IHU care. Most patients receiving both COHC and IHU care were female (43.0%), did not have a spouse (30.6%), had medical assistance (9.2%), had alert consciousness status (79.8%), were aware of end-stage disease (88.3%), had high CCI, and had long stay at the hospice unit (49.6 days).

### 3.2. Hospice Stay and Types of Hospice Care

Table 3 shows the association between the type of hospice care and period or days from the first registration date to death. The period was associated with the type of hospice care (aOR, 1.026; 1.024–1.029, *p* < 0.0001). The odds of receiving both hospice care types increased by 2.6% for one-unit increase in the day of hospice care.

### 3.3. Disease Severity and Type of Hospice Care

Table 4 shows the association between types of hospice care and patient’s disease severity status measured by the CCI. Patients’ disease severity was significantly associated with the type of hospice care (aOR, 1.032; 1.017–1.047, *p* < 0.0001). The odds of receiving both hospice care types increased by 3.2% for one unit increase in the CCI.

### 3.4. Mental Stability, Awareness of End-Stage Disease, and Type of Hospice Care

Table 5 shows the relationship between the types of hospice care and consciousness of patients. Consciousness at the time of hospitalization in the hospice unit was associated with the type of hospice care (aOR, 3.654; 3.269–4.085, *p* < 0.001). The odds of receiving both hospice care types were 3.654 times higher in alert patients compared to those who were not alert.

Table 6 shows the association between the types of hospice care and patient’s status on awareness of terminal illness. Awareness of end-stage diseases was associated with types of hospice care (aOR, 1.422; 1.226–1.650, *p* < 0.0001). The odds of receiving both hospice care types in the group with awareness were 1.422 times higher in patients who were aware compared to those who were not aware of terminal illness.

## 4. Discussion

This study confirmed that more than half of hospice patients used both types of hospice care and that COHC played a critical role in hospice patient delivery systems. The study also found that using hospice services was critically associated with the physical and mental condition of patients: hospice care period from the first evaluation to death, disease severity, consciousness, and awareness of end-stage diseases.

For the percentage of hospice patients receiving both COHC and IHU care, this study found that 58.5% of patients received both COHC and IHU care. This is similar to the study conducted in Netherlands in 2016 stating that 52.4% of hospice users had a history of hospitalization in the hospice care unit [20]. According to a study, the most frequent hospice referral was from the hospital [9]. The study conducted in the United States showed that the hospice utilization rate was 70.8% for patients with cancer and 45.4% for noncancer-related deaths [11]. Accordingly, the COHC implemented in the pilot study for two years made remarkable achievement that greater than half of hospice patients received both hospice programs.

For the period from the first registration to death, this study found that the length of stay in the hospice program was significantly associated with receiving both types of hospice programs. This might be because those receiving both types were initially enrolled to the COHC and then transferred to the IHU. Accordingly, this process may have attributed to the positive relationship between undergoing both programs and period from the first registration to death. Compared to patients who died in the hospital, hospice patients were older, had a shorter length of stay at the hospital, and had more comorbidity [21]. 

For disease severity, this study found that the CCI was positively associated with both types of hospice care received. However, this study result is not aligned with other study results. For example, according to a study conducted in the United States, a low CCI was associated with decreased hospice enrollment [22]. This difference is presumed to occur due to the different pathological conditions of the study subjects such as lung cancer in the study conducted in the United States.

For consciousness at the first registration, this study found that mental consciousness of being alert at the time of the first registration was higher in both users, which presents an opportunity to compare this study results with the previous study findings. Generally speaking, presentation of altered mental status was significantly associated with discharge to hospice care [14]. According to a study targeting patients with primary malignant brain tumors enrolled late in hospice care, a greater proportion of those with late referral were aphasic, unresponsive, and dyspneic compared with patients referred to hospice care more than 7 days before death [13].

For patient’s awareness of end-stage disease, this study found that patients who were aware of end-stage disease were more likely to move to IHU hospice programs or be users of both programs. This study finding is consistent with the result of a previous study in which most patients who received hospice care were aware of their end-stage disease (89.5%) [23]. In Korea, a hospice program was introduced into the national health insurance program two decades ago. Thus, many patients and their family might know that hospice program could alleviate patient’s pain and provide better care at the end-stage of life. Hospice facilities have developed several good programs that many patients with terminal illness want to know [24]. This social demand might motivate patients to be aware of their disease status and select early IHU.

This study has several limitations. First, the hospice registry data from the Korean Hospice Center were duplicative because the center collected information from the patient whenever the patient visited the hospice facility. Depending on the selection of the registration record, the study result may have led to different findings. To minimize possible discrepancies from using different registration records, we used the last record of the patient registry before death. Despite this effort, this might not fully exclude flaws of the study. Second, this study did not include other healthcare utilization of patients. If the data were included, there would be more valid and significant information. Third, when this study recorded both users, it did not differentiate the order of hospice care type use on whether they moved from the IHU to COHC or COHC to IHU. According to unreported research data, most trends or patterns were from COHC to IHU. Accordingly, categorization would lead to different results. 

This study verified that there was high demand on COHC and various patient characteristics affected their stay at COHC or both COHC and IHU. Therefore, if hospitals considered patients’ significant characteristics (period between the first hospice registration and death, disease severity, consciousness status, awareness of end-stage disease) from these study findings during screening for hospice care consultation, more patients who need hospice care will receive sufficient hospice care in a timely manner without unnecessary burdens.

## 5. Conclusions

This study suggests that various patient characteristics are closely related to hospice referral. The study verified that the pilot study project on the COHC had remarkable achievement in that almost 50% of patients had early registration on hospice care and received both COHC and IHU. Both users had different characteristics compared to those using COHC only in standpoints of hospice care period, disease severity, consciousness, and awareness of end-stage diseases. No previous studies had evaluated the government-initiated project on COHC. We hope that the study findings will promote various ideas and insights to effectively utilize hospice care for those with end-stage disease and their families. The study results would also support the government to establish a new policy regarding COHC and to provide ample insight to international colleagues.

## Figures and Tables

**Figure 1 ijerph-19-01566-f001:**
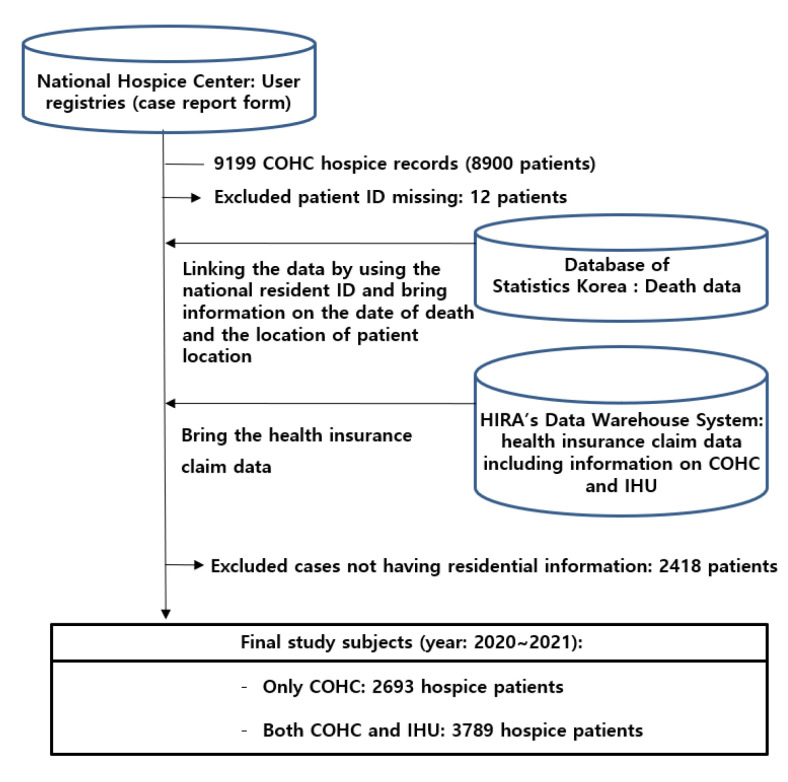
Flow of the data processing procedure.

**Table 1 ijerph-19-01566-t001:** Description of major dependent and independent variables.

Variable	Measures	Source ^a^
Types of hospice	Binary scale: use of COHC only or use of both COHC and IHU	1, 2
Sex	Male versus female	1
Age	Actual age of hospice care patients	1
Marital status	Having a spouse living (no spouse bereavement) versus the others	2
Medical coverage	Health insurance beneficiaries versus national medical assistance	1
Urban/rural ^b^	Urban versus rural areas	3
Main care providers	Having main care providers or not	2
Period between the first hospice registration and death ^c^	Number of days from the first registration to date of death	2
Disease severity ^c^	Using the Deyo method, this study calculated Charlson Comorbidity Index	1
Consciousness ^c^	Mental status of the initial registration	2
Awareness: terminal illness ^c^	Hospice patients were aware of their terminal illness or not	2

^a^ 1, HIRA’s data warehouse system; 2, National Hospice Center; 3, National Statistics Korea; ^b^ urban, an area with a population of more than 100,000; ^c^ Main target independent variables

**Table 2 ijerph-19-01566-t002:** Characteristics of study subjects (*n* = 6482).

Variables	All (*n* = 6482)	COHC Only (*n* = 2693)	Both COHC and IHU (*n* = 3789)	*p* Value
	*n* (%) or M ± SD	*n* (%) or M ± SD	*n* (%) or M ± SD	
Sex				0.0007
Male	3809 (58.8)	1649 (61.2)	2160 (57.0)	
Female	2673 (41.2)	1044 (38.8)	1629 (43.0)	
Age-	66.6 ± 12.6	66.1 ± 12.7	66.9 ± 12.5	0.3212
Marital status				0.0249
Yes (no spouse bereavement)	4567 (70.5)	1938 (72.0)	2629 (69.4)	
No (the others)	1915 (29.5)	755 (28.0)	1160 (30.6)	
Medical coverage				0.0192
Health insurance	5931 (91.5)	2490 (92.5)	3441 (90.8)	
Medical assistance	551 (8.5)	203 (7.5)	348 (9.2)	
Urban/rural				0.4208
Urban	5779 (89.2)	2391 (88.8)	3388 (89.4)	
Rural	703 (10.8)	302 (11.2)	401 (10.6)	
Main care provider				0.6433
Spouse + sibling	5792 (89.4)	2412 (89.6)	3380 (89.2)	
Others	690 (10.6)	281 (10.4)	409 (10.8)	
Period between the first registration and death (days) *	36.2 ± 53.4	17.4 ± 39.1	49.6 ± 58.1	<0.0001
Charlson Comorbidity Index score	10.1 ± 3.4	9.9 ± 3.5	10.3 ± 3.4	<0.0001
Consciousness				<0.0001
Alert	4450 (68.6)	1425 (52.9)	3025 (79.8)	
CNot alert (drowsy, coma)	2032 (31.4)	1268 (47.1)	764 (20.2)	
Awareness of end-stage diseases				<0.0001
Aware	5635 (86.9)	2288 (85.0)	3347 (88.3)	
Not aware	847 (13.1)	405 (15.0)	442 (11.7)	

Note: * days from the first hospice registration to death; M: mean; SD: standard deviation.

**Table 3 ijerph-19-01566-t003:** Factors associated with receiving both hospice care types: days from the first evaluation to death.

Variables	aOR	95% CI		*p* Value
		LL	UL	
Sex: male (Ref = Female)	0.964	0.861	1.078	0.5189
Age	1.005	1.001	1.009	0.0274
Marital status (Ref = No)	0.983	0.861	1.122	0.7999
Medical coverage: HI (Ref = MA)	0.867	0.708	1.061	0.1669
Urban location (Ref = Rural)	0.996	0.842	1.177	0.9593
Main care provider: SS (Ref = The others)	0.967	0.802	1.168	0.7299
Period between the first hospital registration and death	1.026	1.024	1.029	<0.0001

aOR, adjusted odds ratio; HI, health insurance; MA, medical assistance; SS, spouse or sibling; CI, confidence interval; LL, lower limit; UL, upper limit.

**Table 4 ijerph-19-01566-t004:** Factors associated with receiving both hospice care types: CCI.

Variables	aOR	95% CI		*p* Value
		LL	UL	
Sex: male (Ref = Female)	0.837	0.753	0.930	0.0009
Age	1.005	1.001	1.009	0.0142
Marital status (Ref = No)	0.956	0.845	1.083	0.4810
Medical coverage: HI (Ref = MA)	0.819	0.677	0.990	0.0389
Urban location (Ref = Rural)	1.066	0.909	1.249	0.4321
Main care provider: SS (Ref = The others)	1.007	0.843	1.203	0.9375
Charlson Comorbidity Index	1.032	1.017	1.047	<0.0001

aOR, adjusted odds ratio; HI, health insurance; MA, medical assistance; SS, spouse or sibling; CI, confidence interval; LL, lower limit; UL, upper limit.

**Table 5 ijerph-19-01566-t005:** Factors associated with receiving both hospice care types: consciousness.

Variables	aOR	95% CI		*p* Value
		LL	UL	
Sex: male (Ref = Female)	0.849	0.760	0.947	0.0034
Age	1.011	1.007	1.016	<0.0001
Marital status (Ref = No)	0.956	0.839	1.088	0.4925
Medical coverage: HI (Ref = MA)	0.856	0.703	1.042	0.1215
Urban location (Ref = Rural)	1.078	0.914	1.273	0.3725
Main care provider: SS (Ref = The others)	1.082	0.899	1.302	0.4045
Consciousness (Ref = not alert)	3.654	3.269	4.085	<0.0001

aOR, adjusted odds ratio; HI, health insurance; MA, medical assistance; SS, spouse or sibling; CI, confidence interval; LL, lower limit; UL, upper limit.

**Table 6 ijerph-19-01566-t006:** Factors associated with receiving both hospice care types: awareness of terminal illness.

Variables	aOR	95% CI		*p* Value
		LL	UL	
Sex: male (Ref = Female)	0.837	0.753	0.930	0.0009
Age	1.007	1.003	1.011	0.0005
Marital status (Ref = No)	0.947	0.837	1.073	0.3954
Medical coverage: HI (Ref = MA)	0.808	0.669	0.977	0.0278
Urban location (Ref = Rural)	1.052	0.897	1.233	0.5357
Main care provider: SS (Ref = The others)	1.028	0.861	1.229	0.7576
Awareness of end-stage diseases (Ref = No)	1.422	1.226	1.650	<0.0001

aOR, adjusted odds ratio; HI, health insurance; MA, medical assistance; SS, spouse or sibling; CI, confidence interval; LL, lower limit; UL, upper limit.

## Data Availability

The datasets used and/or analyzed in this study are available from the corresponding author on reasonable request.

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
