# Peer review of "Patient Factors Associated with Different Hospice Programs in Korea: Analyzing Healthcare Big Data"

_ijerph, 2022, doi:10.3390/ijerph19031566_

Round 1

Reviewer 1 Report

This is a interesting paper regarding a novel hospice care model initiating in Korea. COHC is very important for terminally-ill patients and it provides opportunities for hospice care team to meet with the patients and family earlier. I believe COHC will be much stronger in Korea, please keep up the work.

However, the study and the manuscript itself has several problems, as below:

Major issues:

  1. The Introduction section was not written in classic way of original article. Normally, we would introduce COHC first, including its history, meaning, and the delivery of care. Then, we would compare COHC to other hospice program, such as hospice inpatient unit or hospice home care, to see the difference and characteristics of COHC, whether in Asian country or in the world. Then we might introduce the COHC in Korea, including why COHC came out at this time? Is there any law related? Then we would tell the readers about what this study aimed for and how this study was conducted. However your introduction was very much alike Discussion section, putting all independent variables into discussion. You also wrote a section of "This study has some significant contributions...." Normally we would put this paragraph in the Discussion section. Please consider a major revision of the Introduction section. 
  2. From Introduction and Discussion, I can not understand why your study design was to compare the characteristics of two groups of hospice patients (IHU and both IHU care and COHC groups). If you want to show the goodness of COHC, it should be "COHC and non-COHC" (Although, you may find it hard to reach the data from your hospice registry database), or "COHC/IHU and IHU alone". It's hard to reveal the characteristics of COHC as long as you used "COHC alone and COHC/IHU", because in cross-sectional design, sometimes patients were too terminally ill to be admitted to IHU and died in acute care unit. 
  3. Likewise, you mentioned "....COHC played a critical role in hospice patient delivery systems..." However it can not be explained from current data, because both groups received COHC. You should consider to re-design the study to show the difference of COHC that you would like to see.
  4. There is also a problem in statistics, in Table 2, there was some differenced between two groups, so you put them in multivariate logistic regression. However, you did not put them altogether but put those independent variables each a time. For example, in Table 3, one unit increase in the CCI would increased the adjusted OR, however consciousness, awareness of disease would simultaneously affect the dependent variables but there were not considered in the model. This could biased how we make conclusions.

Minor issues:

  1. You can put Table 1 into Supplementary Table.
  2. The head of Table 2 is wrong. It should be all participants, COHC, COHC/IHU and p-value.
  3. Extensive English editing should be managed. 
  4. In final paragraph you mentioned "shifted the place of death", however there was no data on place of death in the manuscript.

Author Response

Reviewer 1

This is an interesting paper regarding a novel hospice care model initiating in Korea. COHC is very important for terminally-ill patients and it provides opportunities for hospice care team to meet with the patients and family earlier. I believe COHC will be much stronger in Korea, please keep up the work.

However, the study and the manuscript itself has several problems, as below:

Comment 1:

Major issues:

The Introduction section was not written in classic way of original article. Normally, we would introduce COHC first, including its history, meaning, and the delivery of care. Then, we would compare COHC to other hospice program, such as hospice inpatient unit or hospice home care, to see the difference and characteristics of COHC, whether in Asian country or in the world. Then we might introduce the COHC in Korea, including why COHC came out at this time? Is there any law related? Then we would tell the readers about what this study aimed for and how this study was conducted. However, your introduction was very much alike Discussion section, putting all independent variables into discussion. You also wrote a section of "This study has some significant contributions...." Normally we would put this paragraph in the Discussion section. Please consider a major revision of the Introduction section.

[Our response]

Thank you for your specific suggestion. We explained COHC and the reason why the Korean government introduced the COHC program in the introduction section. This study is to compare the characteristics of patients having only COHC with those having both COHC and IHU. We also added why we are interested in these subjects (please see p.2) in the manuscript. So, we keep the current version of theintroduction section. However, we also appreciate again for your specific suggestions.

We wrote:

 “According to the guideline booklet of the Korean government on the COHC pilot project [8], it was anticipated that the COHC program would help terminally ill patients receive better end-of-life care by registering them to the IHU program earlier. In other words, the Korean government expected the COHC program to serve as the forefront gateway or bridge for early entry into IHU hospice care. This aim was also expected in the pilot project on home hospice care, which was applied as a nationwide program in 2020. According to a study analyzing the pilot project, home hospice care program was effective in early enrollment of patients into the hospice care program [7]. As mentioned above, since COHC was introduced acting and expecting as a bridge to IHU hospice entry, patients using both COHC and IHU would have a longer stay period from entry to hospice to death compared to patients using only COHC. This longer period of stay in hospice programs may suggest that both users be mentally and physically better than those using only IHU program at the time point of hospice enrollment, and as a result, the degree of awareness on their end-stage diseases would be high. However, no study evaluated the effect of the pilot project in these standpoints. Accordingly, it is necessary to evaluate whether early entry into the IHU was achieved and to identify the characteristics of patients who were enrolled in both hospice programs (COHC and IHU) in terms of the enrollment period of a hospice program, disease severity, patient’s consciousness, and their awareness of end-stage disease, in the COHC only and both COHC and IHU..”(p.2)

Comment 2:

From Introduction and Discussion, I can not understand why your study design was to compare the characteristics of two groups of hospice patients (IHU and both IHU care and COHC groups). If you want to show the goodness of COHC, it should be "COHC and non-COHC" (Although, you may find it hard to reach the data from your hospice registry database), or "COHC/IHU and IHU alone". It's hard to reveal the characteristics of COHC as long as you used "COHC alone and COHC/IHU", because in cross-sectional design, sometimes patients were too terminally ill to be admitted to IHU and died in acute care unit.

Likewise, you mentioned "....COHC played a critical role in hospice patient delivery systems..." However it can not be explained from current data, because both groups received COHC. You should consider to re-design the study to show the difference of COHC that you would like to see.

[Our response]

Thank you for the comment. However, if we choose two groups, "COHC and non-COHC", then there still would be many confounding factors among non-COHC. How could we select non-COHC among so many patients? So, we kept the current study design and research model.

Comment 3:

There is also a problem in statistics, in Table 2, there was some differenced between two groups, so you put them in multivariate logistic regression. However, you did not put them altogether but put those independent variables each a time. For example, in Table 3, one unit increase in the CCI would increased the adjusted OR, however consciousness, awareness of disease would simultaneously affect the dependent variables but there were not considered in the model. This could biased how we make conclusions.

[Our response]

Thank you for your comment.We set up separate model because there was high correlation among those variables. So, we believe the current model is appropriate. We also explained it in the method section. Please see p.5. Regarding Odds ratio (OR), it is strictly speaking Adjusted Odds Ratio. For the clarity of the notation, we added a word, “Adjusted”, in each Table.

We wrote:

“Before conducting the main analysis, this study examined the correlations amongthe independent variables to check multicollinearity issue of independent variables. There was a high correlation among the target independent variables, leading the study to establish separate models to consider this effect.” (p.5).

The head of Table 3-6.

Variables

Adjusted OR

95% CI

p-value

Comment 4:

Minor issues:

You can put Table 1 into Supplementary Table.The head of Table 2 is wrong. It should be allparticipants, COHC, COHC/IHU and p-value.Extensive English editing should be managed. In final paragraph you mentioned "shifted the place of death", however there was no data on place of death in the manuscript.

[Our response]

Thank you for your specific suggestion. We thought about putting Table 1 as Supplement Table. However, it is also covenant for readers to read it in the main text. So, we decided to keep it in the main text. Regarding the head of Table 2, we thank you so much for your comment. Since your comment makes sense, we have corrected the first row of Table 2: Variable, All, COHC, Both COHC and IHU, and p-value. For English editing, our manuscript had gotten an English editingservice from a professional editor. As per your last comment, we rewrote the manuscript. 

We wrote:

We corrected the head of Table 2 as follows.

Variables

All

COHC only

Both COHC and IHU

p-value

For wording “shifted the place of death”, we changed the manuscript as “This study verified that there was high demand on COHC and various patient characteristics affected their stay at COHC.”(p.9)

Reviewer 2 Report

  1. Please confirm the content of table 2, the right column looks like “p-value” but the column name is “Max”.
  2. Table 3-6 shows the crude odds ratio of CCI, consciousness, awareness of terminal illness, … etc. How about the adjusted odds ratio of these outcomes of interested?

Author Response

Please confirm the content of table 2, the right column looks like “p-value” but the column name is “Max”.Table 3-6 shows the crude odds ratio of CCI, consciousness, awareness of terminal illness, … etc. How about the adjusted odds ratio of these outcomes of interested?

[Our response]

Thank you for your comments. According to your comment, we corrected the first row of Table 2: Variable, All, COHC, Both COHC and IHU, and p-value. Regarding Table 3-6, out study results were adjusted odds ratio.So, we changed OR to adjusted OR.

We wrote:

We corrected the head of Table 2 as follows.

Variables

All

COHC only

Both COHC and IHU

p-value

For Table 3-6, we changed OR to Adjusted OR. Please see Table 3, 4, 5, and 6.

Reviewer 3 Report

Although there is various literature and background information related to COHC, it is unclear what the author wants to talk about in the introduction. It should contain the specific key ideas for each paragraph.

In the first paragraph of the introduction, the author continues to state that COHC is a new type of program. However, it is still unclear what exact concepts, purpose, or protocol of the COHC program. What is the difference compared to other programs?
the author mentioned that it is called "hospice shared care” or “hospice shared care” in Taiwan... What does mean? It looks the same name... Also, how "shared" care and "consultative" care can be same meaning?
What is the role of COHC? Is this alternation of IHU? pre guiding program of IHU? what is the relationship between IHU and “home-based hospice care”?
The paragraph (line 95) seems not suitable for the introduction. In addition to the past tense, the paragraph is the finding and meaning from the author's study. However, the previous paragraph is only about the rationale and role of hospice care.
The sentence in line 105 "Health insurance claim and...." should be at the method. and "This study has several anticipated policy implications" may be suitable for the discussion section.
In the last sentence "We hope...", the author might be better to state about the research hypothesis instead of expectation.
In figure 1, the shape of the database is distinctive, but the resolution and font are hinder read of the content.
Both "Only COHC" and "both COHC and IHU" are final samples? If so, the author can use these two as block names instead of "Final study subjects". It is confusing. Also, the author can note the number of the hospital for each stage.
Table 1, Having a spouse can be "marital status", Location can be "Urban/rural", State of living with others can be "household members"
Table 2, the first line "title" doesn't match with the following statistics.

The author needs to state the details of the dependent variable. Simple, it can be noted as (COHC only: 0 vs both COHC and IHU: 1). It confuses to interpret later regression results.
Based on the regression table (Table 3~6), each key factor was significantly associated with the odds of both registrations. Here are two considerations of the analysis.
First, these variables such as "consciousness of first registration”, “period between the first hospital registration and death", "aware of end-stage disease" may be associated with the patient's concern of their health condition, and it may lead to the COHC participation. The author already included the patient medical condition at the registration, but how author address the gap among the patient's different levels of knowledge and health concerns?
Second, the causality between registration and outcome variables might be an issue. these variables sequentially happen after the program participation. Therefore, the sample t-test between-group might be a possible choice instead of regression analysis.

Author Response

Comment 1:

Although there is various literature and background information related to COHC, it is unclear what the author wants to talk about in the introduction. It should contain the specific key ideas for each paragraph.

In the first paragraph of the introduction, the author continues to state that COHC is a new type of program. However, it is still unclear what exact concepts, purpose, or protocol of the COHC program. What is the difference compared to other programs?

the author mentioned that it is called "hospice shared care” or “hospice shared care” in Taiwan... What does mean? It looks the same name... Also, how "shared" care and "consultative" care can be same meaning?

What is the role of COHC? Is this alternation of IHU? pre guiding program of IHU? what is the relationship between IHU and “home-based hospice care”?

[Our response]

Thank you for your comments.In the revised manuscript, we wrote “The COHC is a new type of hospice program for terminally ill patients in an acute care unit, which is different from hospice care in an Independent Hospice Unit (IHU). In the COHC, hospice care team is providing hospice care and consultations to patients with terminal illness in acute care units [3, 4]” and hope this clarifies the concept of COHC. We also differentiated two types of hospice care (IHU and home or community-based hospice care) on page 2.

Regarding the duplicated words (“or “hospice shared care””), we deleted one of twobased on your comment.

Comment 2:

The paragraph (line 95) seems not suitable for the introduction. In addition to the past tense, the paragraph is the finding and meaning from the author's study. However, the previous paragraph is only about the rationale and role of hospice care.

[Our response]

Thank you for your comments.SoWe have changed the sentence from the past tense to the future tense (please see p.3).

We wrote:

 “This study is important for several reasons. First, we may be able to recognize social demand on new types of hospice programs in Korea through this study. No previous studies investigated the referral between hospice programs in Korea”(p.3)

Comment 3:

The sentence in line 105 "Health insurance claim and...." should be at the method. and "This study has several anticipated policy implications" may be suitable for the discussion section.

[Our response]

Thank you for your comments. Regarding the manuscript on line 105, we deleted this part because it is already described in the method section. For another comment, we moved this part to the discussion section and slightly rewrote it. (please see p.10)

We wrote:

“This study suggests that various patient characteristics are closely related to hospice referral. The study verified that the pilot study project on the COHC had remarkable achievement in that almost 50% of patients had early registration on hospice care and received both COHC and IHU. Both users had different characteristics compared to those using COHC only in standpoints of hospice care period, disease severity, consciousness, and awareness of end-stage diseases. No previous studies had evaluated the government-initiated project on COHC. We hope that the study findings would promote various ideas and insights to effectively utilize hospice care for those with end-stage disease and their families. The study results would also support the government to establish a new policy regarding COHC and provide ample insight to international colleagues. (p.10)

Comment 4:

In the last sentence "We hope...", the author might be better to state about the research hypothesis instead of expectation.

[Our response]

Thank you for your comments. We used this manuscript for the complement of conclusion. We wrote:

We wrote:

“This study has several anticipated policy implications. This study was conducted to evaluate the government-initiated project on hospice program because no previous studies evaluated the project. We hope that the study findings would promote various ideas and insights to effectively utilize hospice care systems for those with end-stage disease and their families. The study results would support the government to establish a new policy regarding COHC and provide ample insight to international colleagues.” (p.10)

Comment 5:

In figure 1, the shape of the database is distinctive, but the resolution and font are hinder read of the content.

[Our response]

Thank you for your comments. We switched the figure 1 with one having more high resolution.

Comment 6:

Both "Only COHC" and "both COHC and IHU" are final samples? If so, the author can use these two as block names instead of "Final study subjects". It is confusing. Also, the author can note the number of the hospital for each stage.

[Our response]

Thank you for your comments. We changed Figure 1 to reflect the reviewer’s comment (please see Figure 1). However, we did not note the number of the hospitals for each stage because it could make the figure to look complicated and we did not include the number of hospitals in the analysis.

Comment 7:

Table 1, Having a spouse can be "marital status", Location can be "Urban/rural", State of living with others can be "household members"

[Our response]

Thank you for your comments. We changed the location to “Urban/rural”, having a spouse to “Marital status”. However, we deleted “State of living with others” because we did not include this variable in the main analysis.

Comment 8:

Table 2, the first line "title" doesn't match with the following statistics.

[Our response]

Thank you for your comments. Yes, you’re right. We changed the first line title. Thank you again!

Comment 9:

The author needs to state the details of the dependent variable. Simple, it can be noted as (COHC only: 0 vs both COHC and IHU: 1). It confuses to interpret later regression results.

Based on the regression table (Table 3~6), each key factor was significantly associated with the odds of both registrations. Here are two considerations of the analysis.

First, these variables such as "consciousness of first registration”, “period between the first hospital registration and death", "aware of end-stage disease" may be associated with the patient's concern of their health condition, and it may lead to the COHC participation. The author already included the patient medical condition at the registration, but how author address the gap among the patient's different levels of knowledge and health concerns?

Second, the causality between registration and outcome variables might be an issue. these variables sequentially happen after the program participation. Therefore, the sample t-test between-group might be a possible choice instead of regression analysis.

[Our response]

Thank you for your comments. Yes, you’re right. We changed the first line title. Thank you again! Regarding the two considerations, we already explained the reason why we set up the separate models in the method section. For clarity of the reason, we slightly emended the manuscript text. Please see the manuscript on p.5.

We wrote:

“The dependent variable was a binary scale: COHC only or both COHC and IHU (COHC only: 0 vs both COHC and IHU: 1)” (p.5)

“Before conducting the main analysis, this study examined the correlations among the in-dependent variables to check multicollinearity issue of independent variables. There was a high correlation among the target independent variables, leading the study to establish separate models to consider this effect.” (p.5)

Reviewer 4 Report

Thank you for the opportunity to review this manuscript. This was an interesting and well-written paper, addressing an important topic. I do, however, have some concerns and questions and some relatively minor suggestions for shaping the manuscript up for publication:

1- In the introduction they indicate that   “it is necessary to evaluate whether early entry into the IHU 57 was achieved and identify the characteristics of patients who were enrolled in both hospice programs (COHC and IHU) in terms of disease severity, patient's consciousness, awareness of end-stage disease, and enrollment period of a hospice program, in the COHC only and both COHC and IHU ”(lines 57-60).   However, they do not clarify why they choose these variables. Perhaps these variables are related to criteria for referral or inclusion of patients to COHC and IHU? It is important to know this data, to better understand the conclusion: “This study suggests that various patient characteristics are closely related to hospice referral” (lines 292-293).  It is not the same that these characteristics are criteria prior to referral than referral outcome variables.

2- In the Results, table 2 is poorly constructed; surely a column has been altered and the data does not match the columns, according to the text that describes that table. They must correct it

3-The discussion is very well organized, it is quite clear and easy to read, technically correct, however in general it is too descriptive, it lacks involvement and the point of view of the authors. Contributing your point of view can add strength to what you discuss.

Likewise, there are some inaccuracies regarding the literature consulted that should be specified in order to improve the quality of the discussion. For example, they say: “For disease severity, this study found that the CCI was positively associated with receiving both types of hospice care, which is aligned with other study results. According to a study conducted in the United States, a low CCI was associated with decreased hospice enrollment [21] " (lines 241-244).   But this happened for lung cancer patients, it should not be expressed in a general way since this study to which they refer deals only with lung cancer patients and in their study they have not detailed the pathologies. The correct thing is to indicate that in studies of specific pathologies such as lung cancer also a low CCI was associated with decreased hospice enrollment.

Something similar happens in the next paragraph.

They probably have not found similar published studies, so the discussion should be made with articles with similar or opposite results, but that does not mean that it is not specified that they are different studies from yours.

I encourage you to make these small modifications with the intention of improving it for publication.

Author Response

Thank you for the opportunity to review this manuscript. This was an interesting and well-written paper, addressing an important topic. I do, however, have some concerns and questions and some relatively minor suggestions for shaping the manuscript up for publication:

Comment 1:

1- In the introduction they indicate that “it is necessary to evaluate whether early entry into the IHUwas achieved and identify the characteristics of patients who were enrolled in both hospice programs (COHC and IHU) in terms of disease severity, patient's consciousness, awareness of end-stage disease, and enrollment period of a hospice program, in the COHC only and both COHC and IHU”(lines 57-60). However, they do not clarify why they choose these variables. Perhaps these variables are related to criteria for referral or inclusion of patients to COHC and IHU? It is important to know this data, to better understand the conclusion: “This study suggests that various patient characteristics are closely related to hospice referral” (lines 292-293).  It is not the same that these characteristics are criteria prior to referral than referral outcome variables.

[Our response]

Thank you for your comments. You did point out the weakness of our manuscript. To reflect your comments, we added several our thoughts and expectations. In addition, we reordered the contents of topics with hospice stay period followed by CCI, consciousness, and awareness of end-stage diseases. Please see the manuscript on page 2 and Table 2-6.

We wrote:

 “As mentioned above, since COHC was as a bridge to IHU hospice entry, patients using both COHC and IHU would have a longer stay period from entry to hospice to death compared to patients using only COHC. This longer period of stay in hospice care pro-grams may suggest that both users be mentally and physically better than those using only IHU program at the time of hospice enrollment, and as a result, the degree of awareness on their end-stage diseases would be high. However, no study evaluated the effect of the pilot project in these standpoints. Accordingly, it is necessary to evaluate whether early entry into the IHU was achieved. Also it is valuable to identify the characteristics of patients who were enrolled in both hospice programs (COHC and IHU) in terms of the enrollment period of a hospice program, disease severity, patient’s consciousness, and their awareness of end-stage disease, in the COHC only and both COHC and IHU.” (p.2) (please see Table 2-6.)

Comment 2:

2- In the Results, table 2 is poorly constructed; surely a column has been altered and the data does not match the columns, according to the text that describes that table. They must correct it

[Our response]

Thank you for your comments. You’re right. We accept your comment. So, we corrected the first row of Table 2: Variable, All, COHC, Both COHC and IHU, and p-value. Regarding Table 3-6, out study results were adjusted odds ratio. So, we changed OR to adjusted OR.

We wrote:

We corrected the head of Table 2 as follows.

Variables

All

COHC only

Both COHC and IHU

p-value

Comment 3:

3-The discussion is very well organized, it is quite clear and easy to read, technically correct, however in general it is too descriptive, it lacks involvement and the point of view of the authors. Contributing your point of view can add strength to what you discuss.

Likewise, there are some inaccuracies regarding the literature consulted that should be specified in order to improve the quality of the discussion. For example, they say: “For disease severity, this study found that the CCI was positively associated with receiving both types of hospice care, which is aligned with other study results. According to a study conducted in the United States, a low CCI was associated with decreased hospice enrollment [21] " (lines 241-244).   But this happened for lung cancer patients, it should not be expressed in a general way since this study to which they refer deals only with lung cancer patients and in their study they have not detailed the pathologies. The correct thing is to indicate that in studies of specific pathologies such as lung cancer also a low CCI was associated with decreased hospice enrollment.

[Our response]

Thank you for your specific comments. You are really pointing out our lack of involvement and inaccurate discussion. So, we rewrote the manuscript. Please see the manuscript on p.8

We wrote:

"For disease severity, this study found that the CCI was positively associated with receiving both types of hospice care. However, this study result is not aligned with other study result. For example, according to a study conducted in the United States, a low CCI was associated with decreased hospice enrollment [22]. This difference is presumed to occur due to the different pathological conditions of the study subjects such as lung cancer in the study conducted in the United States."(p.8)

Comment 3-1:

Something similar happens in the next paragraph.

They probably have not found similar published studies, so the discussion should be made with articles with similar or opposite results, but that does not mean that it is not specified that they are different studies from yours.

[Our response]

Thank you for your comments. We changed the manuscript with a little bit low tone. Please see the manuscript on p.8

We wrote:

"For consciousness at the first registration, this study found that mental consciousness of being alert at the time of the first registration was higher in both users, which presents an opportunity to compare this study results with the previous study findings."(p.8)

I encourage you to make these small modifications with the intention of improving it for publication.

[Our response]

Thank you so much for all your specific comments. We deeply appreciate them. Your comments were tremendously helpful for us to improve our manuscript. Thank you so much again!

Round 2

Reviewer 1 Report

Thanks to authors for the revised manuscript. It is clearly written, and most of my previous questions have been answered properly. However the paper would be greatly strengthened and become an important piece of research internationally if implications for COHC can be more emphasized. 

I note that the average contact time for COHC was 17.4 days. Whats the average contact time for patients with IHU only? Please state how these “terminal” patients were recruited to COHC service, and whether you think many palliative care service rather later than they might fully benefit. Please comment if there might be an opportunity to start a palliative care approach earlier in hospital.

Finally, might the COHC has a teaching and training role in the hospital to train hospital doctors to provide generalist palliative care to patients before they are terminally ill?  Considering these points will raise the status of the paper from a service evaluation to a research paper with implications for service provision, training and further research.

Author Response

Response to Reviewer 1 Comments

Point 1: Thanks to authors for the revised manuscript. It is clearly written, and most of my previous questions have been answered properly. However the paper would be greatly strengthened and become an important piece of research internationally if implications for COHC can be more emphasized.

Response 1: Thank you for your comments. We deeply appreciate your encouragement and compliments. We already included some manuscript on implications for COHC. So, we did not include any further manuscript. Please see the beginning of the discussion section.

Point 2: I note that the average contact time for COHC was 17.4 days. What is the average contact time for patients with IHU only? Please state how these “terminal” patients were recruited to COHC service, and whether you think many palliative care service rather later than they might fully benefit. Please comment if there might be an opportunity to start a palliative care approach earlier in hospital.

Response 2: Thank you for your question. We have a reference on ALOS of patients for IHU users only. ALOS of patients of IHU was 20.0 days for tertiary hospitals, 26.0 days for general hospitals, 27.3 days for small hospitals, 30.4 days for long-term care hospitals, and 25.9 days for some clinics. These types of hospitals had IHU (independent hospice unit). So, we concluded that COHC plays a critical role in hospice patient delivery systems bringing more patients with end-stage diseases into hospice programs. We already described this fact in the manuscript and, thus, did not include manuscript anymore regarding this subject. For your question regarding the opportunity to start a palliative care approach earlier in hospital, we think that COHC programs is one of those starting palliative cares in hospital wards, but not independent hospice unit(ward). If a patient is diagnosed with terminal illness and consents to get hospice cares, they may be eligible for COHC.

Point 3: Finally, might the COHC has a teaching and training role in the hospital to train hospital doctors to provide generalist palliative care to patients before they are terminally ill?  Considering these points will raise the status of the paper from a service evaluation to a research paper with implications for service provision, training and further research.

Response 3: Thank you for your question. Yes, we have. Medical doctors and nurses who are participating in COHC programs should have various palliative care training programs and continuous education (CE) on all types of hospice programs annually, which are mandatory programs. These doctors and nurses are teaching and training the medical staffs (doctors and nurses) working at acute care wards for patients with terminal illness. These doctors and nurses are also providing palliative care and hospice care in general acute care wards.

Reviewer 2 Report

1. Variables in Table2 contain continuous and categorical variables, the content needs to present what is mean(std) and what is n(%). Besides, the abbreviation of Table2 needs consistency.
2. Results in Table3-6 were adjusted odds ratio (aOR), the section of Reults should use "aOR" in line 201, line 208, line 215, and line 222.

Author Response

Response to Reviewer 2 Comments

Point 1: Variables in Table2 contain continuous and categorical variables, the content needs to present what is mean(std) and what is n(%). Besides, the abbreviation of Table2 needs consistency.

Response 1: Thank you for your comments. Yes, you’re right. So, we put “n(%)” for categorical variable and n(SD) for numeric variable. For consistency, we used “SD” for standard deviation, but not use “std”. Please see Table 2.

Point 2: Results in Table3-6 were adjusted odds ratio (aOR), the section of Reults should use "aOR" in line 201, line 208, line 215, and line 222.

Response 2: Thank you for your specific comments. We changed all the manuscript using oOR following your comments. Thank you again!

Reviewer 3 Report

This sentence doesn’t clear – “This study results indicate that referrals from COHC to IHU “meaning bother users” may be related to patients’ mental status (line 93)”

Regarding comment 2, the paragraph of research purpose (line 111) should be supported. The following components will be helpful: (1) major characteristics of research, (2) research hypothesis, (3) methodology and approaches, or (4) a brief overview of the structure of the paper.

As I mentioned in comment 2, the updated paragraph (p.10) for comment 4 is the same thing as the importance of research in the introduction (line 103).

Regarding comment 6, I am still curious how many hospitals were included in the research. Is there any difference of hospital between COHC only and both COHC and IHU programs? Since I believe that the quality of the program might be varying within each hospital, I am wondering how an author can control this difference.

Table 2 still needs to be improved. The author can add “n (%)” on the second row and also note “m (sd)” for the related row (e.g., age).

Author Response

Response to Reviewer 3 Comments

Point 1: This sentence doesn’t clear – “This study results indicate that referrals from COHC to IHU “meaning bother users” may be related to patients’ mental status (line 93)”

Response 1: Thank you for your good comment. You are right. There was a typo. We changed the words from “meaning bother users” to “meaning both users”. Please see the manuscript in p.3.

Point 2: Regarding comment 2, the paragraph of research purpose (line 111) should be supported. The following components will be helpful: (1) major characteristics of research, (2) research hypothesis, (3) methodology and approaches, or (4) a brief overview of the structure of the paper.

Response 2: Thank you for your comments. We rewrote the study purpose to make it clearer and to reflect your comments, which focuses on the investigation of the relationships between two types of hospice use and the four target independent variables. Please see the end of the introduction section. We wrote as “This study aimed to investigate the relationships between the use of different hospice programs and the four characteristics of hospice care recipients: the enrollment period of hospice program, disease severity, patient’s consciousness, and their awareness of end-stage diseases.”(p.3).

Regarding your 3 comments, we put some texts in the manuscript. We added some texts as “For mental and physical status of patients, this study selected three factors: disease severity, patient’s consciousness, and their awareness of end-stage disease. Investigating the relationships between use of different hospice programs and these factors is one of major features of this study. However, no previous study evaluated its relationships of the pilot project in these standpoints. Accordingly, it is necessary to evaluate whether early entry into the IHU was achieved and whether there were any relationships between use of different hospice programs such as COHC only or both hospice programs (COHC and IHU) and characteristics of patients in the standpoint of disease severity, patient’s consciousness, and their awareness of end-stage disease. This study hypothesized that hospice recipients who use both would have longer stay in hospice program, better physical and mental status compared to those of recipients with COHC only. For readers’ better understanding, this study constructed the presentation of the study results in the order of patients’ enrollment period of hospice program which is the main purpose of pilot project followed by disease severity, patient’s consciousness, and their awareness of end-stage disease.”(p.2).

Point 3: As I mentioned in comment 2, the updated paragraph (p.10) for comment 4 is the same thing as the importance of research in the introduction (line 103).

Response 3: Thank you for your comments. We agree with you. So, we deleted some parts of manuscript located in the introduction. Please see the second paragraph at the end of the introduction section.

Point 4: Regarding comment 6, I am still curious how many hospitals were included in the research. Is there any difference of hospital between COHC only and both COHC and IHU programs? Since I believe that the quality of the program might be varying within each hospital, I am wondering how an author can control this difference.

Response 4: Thank you for your comments. We understand that your curiosity is reasonable and we should have put the information in the manuscript. For hospitals running hospice programs focusing on COHC and IHU (excluding Home Hospice Care Program), there are 9 hospitals running only COHC (T1), 18 hospitals running both COHC and IHU programs (T2), and 70 hospitals running only IHU (T3). So, patients can be transferred to T3 from T1 or within T2. This study also considered these patients as a group using both COHC and IHU. There are complicated types of referrals such as one mentioned above. So, we simplified the types of referrals with two scales: (COHC only: 0 versus both COHC and IHU: 1). For readers’ understanding, we put this explanation into the manuscript, especially in the method section (study design section, p.3). For your comments regarding the controlling the difference, we did not control its effects because there were various different types of referrals.  

Point 5: Table 2 still needs to be improved. The author can add “n (%)” on the second row and also note “m (sd)” for the related row (e.g., age).

Response 5: Thank you for your comments. You’re right. So, we put “n(%)” for categorical variable and n(SD) for numeric variable. For consistency, we used “SD” for standard deviation, but not use “std”. Please see Table 2.
